# Gut metagenome associations with extensive digital health data in a volunteer-based Estonian microbiome cohort

Oliver Aasmets[1,3], Kertu Liis Krigul[1,3], Kreete Lüll [1], Andres Metspalu [1,2] & Elin Org [1✉]

Microbiome research is starting to move beyond the exploratory phase towards interventional trials and therefore well-characterized cohorts will be instrumental for generating hypotheses and providing new knowledge. As part of the Estonian Biobank, we established the Estonian Microbiome Cohort which includes stool, oral and plasma samples from 2509 participants and is supplemented with multi-omic measurements, questionnaires, and regular linkages to national electronic health records. Here we analyze stool data from deep metagenomic sequencing together with rich phenotyping, including 71 diseases, 136 medications, 21 dietary questions, 5 medical procedures, and 19 other factors. We identify numerous relationships ($n = 3262$) with different microbiome features. In this study, we extend the understanding of microbiome-host interactions using electronic health data and show that long-term antibiotic usage, independent from recent administration, has a significant impact on the microbiome composition, partly explaining the common associations between diseases.

[1] Estonian Genome Centre, Institute of Genomics, University of Tartu, Tartu, Estonia. [2] Institute of Cell and Molecular Biology, University of Tartu, Tartu, Estonia. [3] These authors contributed equally: Oliver Aasmets, Kertu Liis Krigul. ✉email: elin.org@ut.ee

 1

In the last decade, human microbiome research has undoubtedly been one of the most exciting and most rapidly expanding fields from the perspective of personalized medicine. Technological developments and increasing amounts of available data have transformed microbiome science, expanding our knowledge about the ways the microbiome is influenced by our lifestyle choices and the environment we live in as well as how it reflects the state of our health. In addition, the mechanisms by which unfavorable changes in the microbiome might lead to the deterioration of our well-being and the ways that the microbiome could be altered to interfere with the progression of disease are being studied[1,2]. The focus of precision health research has recently moved beyond analyzing host DNA to study host-related factors, including among others the microbial communities[3,4]. Thus, population-scale studies of the microbiome and health relationships are of utmost importance.

Recently, Wilkinson et al. have published a comprehensive perspective on incorporating the current knowledge about the human microbiome into population-scale health research and practice[5]. The authors highlight several areas of microbiome science that can have a major impact on public health, including the identification of microbiome–drug interactions that are responsible for dose effectiveness and adverse events, management of chronic diseases, and maintenance of one's health. Biobanks can help to achieve these goals, since they include large sample sets of multiple data layers and provide excellent metadata from anthropometric measurements, questionnaires, and health registries.

Although the gut microbiome profiles and factors shaping the community dynamics and function have been described in various populations, only a few extensive population-based metagenomic cohorts are available[6–8]. The first large-scale population level association studies have extended our knowledge regarding the relative impact of various host and environmental factors on the composition of the microbiome[6]. Moreover, a recent extended study of 8208 individuals in the Dutch population shows that the 241 studied phenotypes from a broad set of categories explained approximately 15% of the variation in the microbiome composition and function between individuals, indicating that our current understanding of the factors that shape the microbiome is still limited[7]. Thus, prospective biobanks with extensive lifestyle and health data from different populations will be an important resource for dissecting the role of the gut microbiome in the near future.

The Estonian Biobank (EstBB) is one of the leading volunteer-based biobanks in Europe, covering almost 20% of the country's adult population (> 200,000 participants ≥ 18 years old). Additionally, comprehensive phenotype data are available from the nationwide electronic health records (EHRs), enabling monitoring of the health status across a lifespan. The EHRs are recorded by medical specialists, thus providing reliable information for disease diagnosis and prescribed medications. In this study, we analyzed the gut microbiome composition in more than 2500 adults from the EstBB who participated in the Estonian Microbiome Project (EstMB), which included stool sample collection and comprehensive phenotyping. The project was established in order to study the role of the microbiome in health and disease as well as to evaluate its potential as a means to advance personalized medicine.

In this work we show that by using deep metagenomic sequencing, we are able to identify the impact of 50 diseases, 47 medications, 20 dietary factors, 5 intrinsic factors, 4 medical procedures, and 14 other factors on the gut microbiota, which together explain 10.14% of the inter-individual variation in the gut microbiota composition and 10.48% of the variation in the microbiome functionality. Access to EHRs also provided us with a unique opportunity to study the effect of long-term medication usage on the fecal microbiome. We demonstrate that the long-term use of antibiotics has a remarkable effect on microbiome diversity and might partly explain shared dysbiosis between diseases. After correcting for the number of antibiotic treatments taken over the last 10 years, we identify a clear decline in the number of previously detected microbiome-disease associations, underlining the value of longitudinal health data records in interpreting the results and identifying disease-specific signals. The EstMB cohort is an excellent resource for analyzing the role of fecal microbiota in disease susceptibility, clinical phenotypes, and therapeutic responses using information on past and future clinical outcomes by linkage to the participants' EHRs.

## Results

**Cohort overview.** The EstBB is a volunteer-based cohort of the Estonian adult population initiated in 1999 with the objective to investigate the genetic, environmental, and behavioral background of common diseases and traits[9]. Currently, the EstBB includes more than 200,000 genotyped participants (≥ 18 years old) from all over the country. Supported by the Estonian Human Genes Research Act, continuously updated personal health data from various EHRs and re-examinations as well as a wide array of biological samples are used to promote the development of personalized medicine and to improve public health[9]. To identify and characterize microbiome-associated factors, the Estonian Microbiome project was initiated in 2017. More than 2500 EstBB participants, who joined the EstBB at least 10 years ago, provided stool, oral, and blood samples to establish the EstMB cohort (Fig. 1a). Being part of the larger EstBB cohort, all of the EstMB participants have been genotyped, and additional omics data (whole-exome sequencing, whole-genome sequencing, metabolomics, etc.) are available for subsets of participants (Supplementary Fig. 1). Additionally, all of the EstBB and EstMB participants have responded to extensive questionnaires covering their dietary preferences, living environment, and various lifestyle choices. The overview of the EstMB cohort and data characteristics is shown in Fig. 1.

The EstMB cohort includes 2509 adult participants (age range: 23–89 years old, mean: 50.1 ± 14.93 years old) of whom 70.3% are women ($n = 1764$) and 29.7% are men ($n = 745$) [body mass index (BMI) range: 15.1–54.0 kg/m$^2$, mean: 26.5 ± 5.34 kg/m$^2$)] (Fig. 1b, Supplementary Table 1). The vast majority of the participants are of Estonian origin (98.4%). The majority of the participants originate from two counties, Tartu County (28.3%) and Harju County (25.3%), where most of the Estonian population is located; however, all 15 Estonian counties are represented (Fig. 1c).

One of the major strengths of the EstBB is the possibility to use EHRs for investigating medical procedures, disease occurrences, and medication usage before and after sample collection and to recontact the subjects in the biobank. This enabled us to study the effects of disease phenotypes, medication usage, and disease–drug interactions on the microbiome in great detail. For example, Fig. 1d illustrates the value of using EHRs compared to self-reported questionnaires for identifying disease cases. Besides detecting only a small proportion of cases recorded in EHRs, questionnaires fail to cover the vast majority of total disease counts. Questionnaires can occasionally provide additional information to EHRs; however, the reliability of the self-reported diagnosis is in some cases questionable (e.g., Crohn's disease and depression). Therefore, we focused primarily on the EHRs for the identification of prevalent diseases and medications used.

**Landscape of the Estonian gut microbiome.** To characterize the composition and functionality of the microbiome in the Estonian

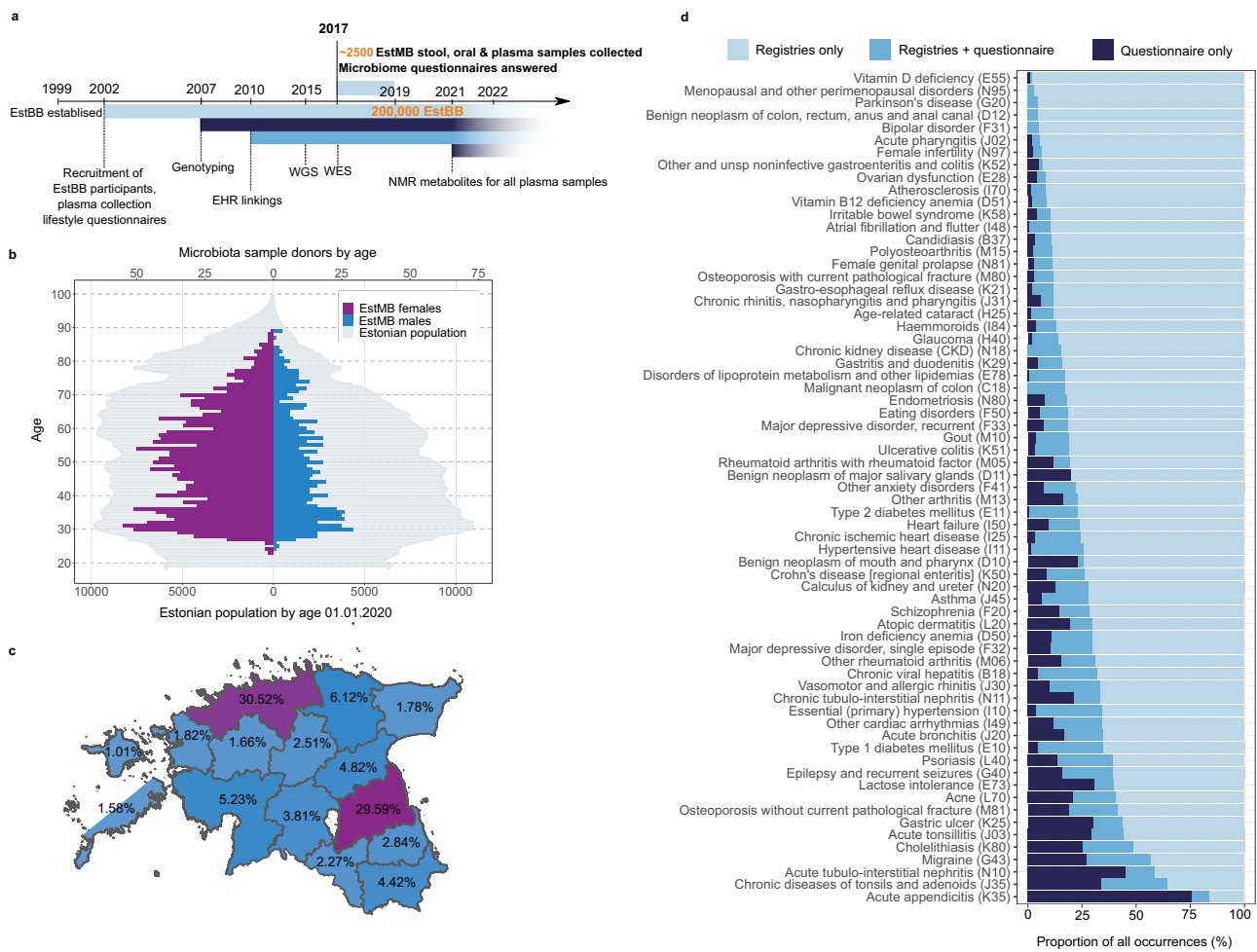

**Fig. 1 EstMB cohort characteristics. a** Data flow of the EstMB cohort participants. In 2009–2010 and during the stool sample collection in 2017–2019, subjects (*N* = 2509) completed questionnaires on their diet and lifestyle. All of the EstMB participants have been genotyped. Additional datasets, e.g., WES, WGS, and NMR metabolite datasets, are available for subsets of participants (Supplementary Fig. 1). Data on disease occurrences, prescribed medications, and medical procedures have been recorded annually for all of the participants using the national EHRs. **b** The age and gender distributions of the cohort participants were compared to the Estonian population in 2020. **c** Residence of the cohort participants by county. The two counties colored with dark purple are where most of the population originates from and where the study centers were located. The figure uses county borders data from the Estonian Land Board (accessed 12.01.2021). **d** Compliance of prevalent diseases from EHRs and self-reported questionnaires. The color bars indicate the data source of the diagnosis (light blue, registries only; blue, registries and questionnaire; dark blue, questionnaire only). EstMB Estonian Microbiome Project, EHRs Electronic health records, NMR Nuclear magnetic resonance, WGS Whole-genome sequencing, WES Whole-exome sequencing.

population, the microbial DNA was extracted using a QIAamp DNA Stool Mini Kit (Qiagen, Germany) and then sequenced using paired-end metagenomic shotgun sequencing on the Illumina Novaseq 6000 platform, resulting in an average of 4.62 ± 0.44 Gb of data per sample. The mean relative abundance of bacterial taxa was 98.51%, followed by 0.55% for taxa of viral origin, 0.35% for taxa of eukaryotic origin, and 0.08% for taxa belonging to the Archaeal kingdom (Supplementary Table 2).

The Estonian gut microbiome community and functional profiles are characterized in Fig. 2. Similar to previously reported results, the microbiome taxonomic composition is highly variable across the population, while the functional profile is stable (Fig. 2a, b)[7]. The Bacteroidetes and Firmicutes phyla dominate the taxonomic composition, with average relative abundances of 56.2% (range: 0.2–95.2%) and 33.7% (range: 1.8–91.6%), respectively, followed by Proteobacteria with 5.4% (range: 0.8–93.6%) and Actinobacteria with 1.0% (range: 0.1–10.6%) (Fig. 2a, Supplementary Table 2). Bacteroidetes and Firmicutes together account for a similar percentage of the whole microbiome composition in the EstMB compared to the largest current

European microbiome population study, the Dutch Microbiome Project (DMP) (~90% on average in the EstMB vs. ~83% in the DMP)[7], although the relative abundance of Firmicutes is higher in the Estonian population (33.7% in the EstMB vs. 24.8% in the DMP). The most dominant genera in the samples correspond to the previously reported enterotypes, with *Bacteroides* and *Prevotella* genera being the most dominant followed by the genus *Clostridium* (Fig. 2c)[10].

To characterize the microbial taxa shared by most of the Estonian population, we studied the core genera of the cohort using a similar definition as the Finnish FINRISK cohort[8]. In particular, we defined the core genera as those with a relative abundance of > 0.1% in at least 10% of the samples. In total, we identified 72 genera as the core, of which 70 were bacterial genera, 1 was eukaryotic (*Blastocystis*), and 1 was viral (*Sk1virus*) (Supplementary Table 2). The mean relative abundance of the identified core was 84.4%. The identified core genera are similar to results reported previously. Among the 72 identified genera, 43 were shared with the Finnish FINRISK cohort core[8]. In addition, all of the 9 reported core genera in the large 16 S rRNA based

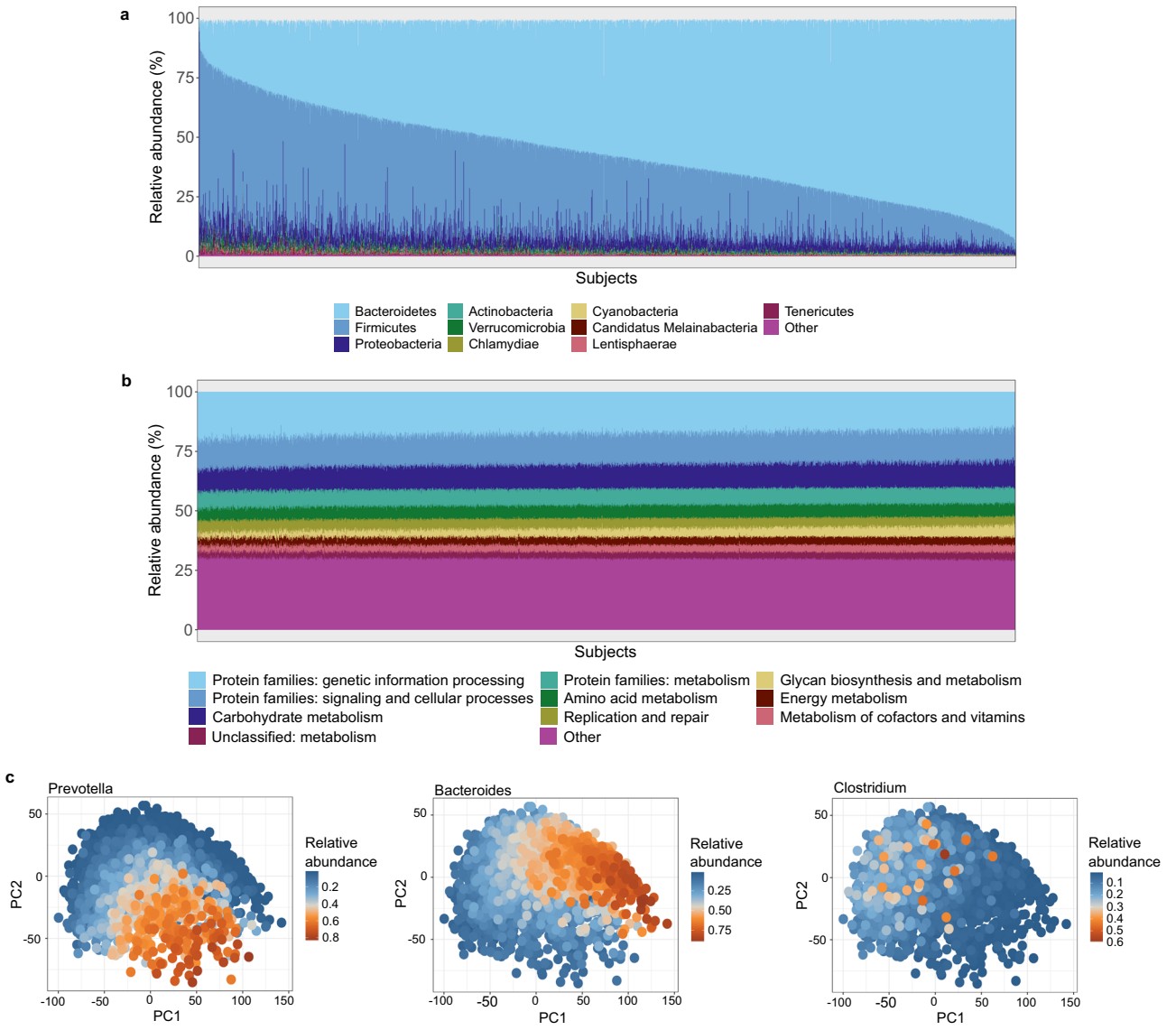

**Fig. 2 Landscape of the Estonian microbiome. a** Phylum level microbiome composition across all EstMB cohort subjects. **b** Functional profile of the microbiome across EstMB cohort subjects (KEGG domains). **c** PCA biplots on species-level taxonomic profiles colored by the relative abundance of the most dominant genus in the sample. EstMB Estonian Microbiome Project, KEGG Kyoto Encyclopedia of Genes and Genomes, PCA Principal component analysis.

MiBioGen consortium including 24 cohorts were also present in the EstMB core[11].

**Overview of microbiome–phenotype associations**. Next, we aimed to get an extensive look at the microbiome-associated factors in the Estonian population. A total of 252 factors were analyzed, including 71 prevalent diseases, usage of 136 medications, 21 factors reflecting dietary preferences, 5 medical procedures, and 19 other factors describing lifestyle and anthropometric measurements (Supplementary Table 1). To determine which factors are associated with the overall microbiome composition, we analyzed the association of each factor with the microbial alpha diversity (observed richness and Shannon's index) and beta diversity [interindividual differences in the microbiome compositions calculated based on the Euclidean distance on centered log-ratio (CLR)-transformed data on 17,158 species and 7869 Kyoto Encyclopedia of Genes and Genomes (KEGG) orthologies (KOs). Figure 3 shows the statistically significant associations (false

discovery rate, FDR < 0.05) with a species-level microbiome profile. In total, 39 factors were associated with the species richness, i.e., observed number of unique species, 20 factors were associated with the Shannon's index, and 75 factors were found to be associated with the beta diversity calculated on the species-level profile. Diversity analyses on the functional profile (KO profile) showed associations between 16 factors and the KO richness, and 90 factors were found to be associated with the beta diversity calculated on the KO profile (Supplementary Fig. 2, Supplementary Tables 3 and 4).

The microbial alpha diversity, characterized by the number of observed species and the Shannon entropy, confirmed the previously reported associations with microbial diversity. Stool characteristics such as consistency (higher Bristol scale values), medication usage, and disease prevalence were associated with a lower biodiversity as previously reported[6,12,13]. Other lifestyle factors reflecting the so-called healthy lifestyle, such as physical exercise and consumption of berries, fruit, and vegetables, corresponded to a higher diversity; whereas smoking, a higher

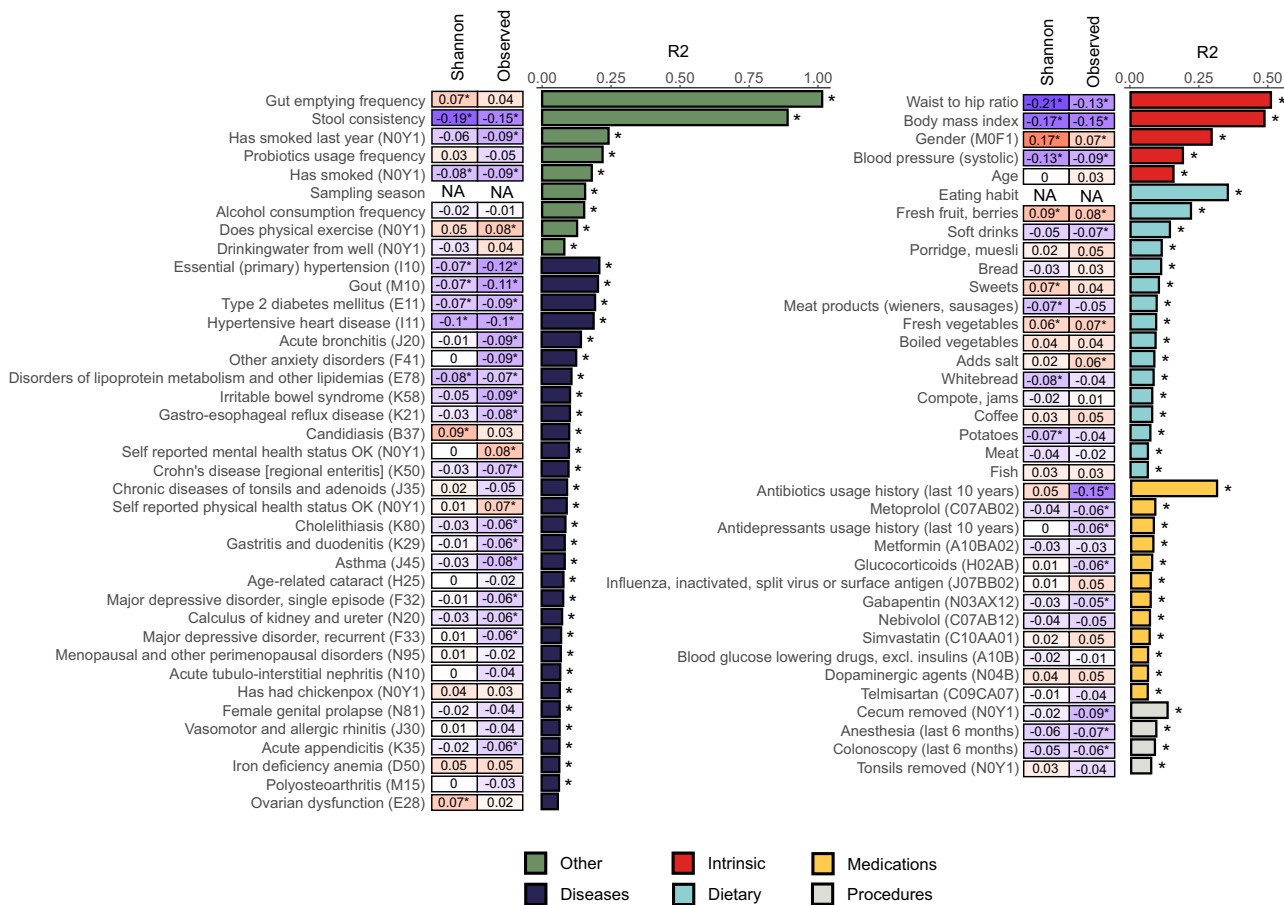

**Fig. 3 Statistically significant associations with species-level microbiome alpha and beta diversity.** The bar plot indicates the explained variance in the interindividual variation of the microbial composition obtained by the permutational analysis of variance (based on the Euclidean distance on the centered log-ratio-transformed data). The heatmap shows the Spearman correlation coefficients of each factor with the Shannon's index of diversity and the observed species richness. Blue indicates a negative correlation, and red indicates a positive correlation. Asterisks indicate associations with false discovery rate < 0.05. In the brackets are the international classification of diseases-10 and anatomical therapeutic chemical codes for diagnoses and medications, respectively.

BMI, and the consumption of soft drinks and preprocessed meat products were associated with a lower diversity (Fig. 3, Supplementary Fig. 2, Supplementary Table 3). The strongest associations with the beta diversity (FDR < 0.001), both with specific species and pathways, were observed for the gut emptying frequency ($R^2 = 1.03\%$), stool consistency (Bristol stool scale, $R^2 = 0.95\%$), waist-to-hip ratio ($R^2 = 0.52\%$), and BMI ($R^2 = 0.49\%$) (Fig. 3, Supplementary Fig. 2, Supplementary Table 4). The diseases describing the most variation in the microbiome composition (FDR < 0.001) included hypertension ($R^2 = 0.43\%$ and $R^2 = 0.21\%$, for pathways and species, respectively), gout ($R^2 = 0.18\%$ and $R^2 = 0.21\%$), and type 2 diabetes (T2D) ($R^2 = 1.1\%$ and $R^2 = 0.20\%$) (Fig. 3, Supplementary Fig. 2, Supplementary Table 4).

In line with previous studies[6,14], several drugs were significantly associated with the microbiome composition, of which the beta blocker metoprolol, metformin, and glucocorticoids described the most variation amongst the medications used (Fig. 3, Supplementary Table 4). Using the EHR data, we were able to expand the current knowledge by studying and comparing the effects of drugs belonging to the same pharmacological or chemical subgroups [up to Anatomical Therapeutic Chemical (ATC) classification level 5]. For example, the beta blockers metoprolol and nebivolol were found to be associated with the gut microbiome, with metoprolol describing more variation in the gut microbiome (Supplementary Table 4).

In addition to replicating the findings from previous studies, our study yielded novel associations. Our results expand the knowledge regarding microbiome-related factors by showing novel associations with recently carried out medical procedures (anesthesia and colonoscopy), dietary factors (drinking water origin), and usage of various medications [e.g., glucocorticoids (H02AB); simvastatin (C10AA01); dopaminergic agents (N04B); blood glucose-lowering drugs, excl. insulins (A10B); and influenza, inactivated, split virus or surface antigen (J07BB02) (Supplementary Table 4)]. Altogether, the studied factors describe 10.14% of the total variability in the species-level microbial composition and 10.50% of the variability in the KO profile.

To obtain a more precise understanding of the associations seen in the overall microbial composition, unique species and KOs were associated with the diseases, medications, and dietary factors using the ALDEx2 methodology after adjusting for gender, age, BMI, and stool type (Methods). In order to limit the number of comparisons, the species and KOs that were detected with > 10% prevalence at a relative abundance of 0.1% were used for the analysis, resulting in 1231 species and 1974 KOs. In total, we identified 1335 statistically significant associations with the species and 1072 associations with the KOs (FDR < 0.05) (Supplementary Table 5).

In summary, 50 diseases, usage of 47 medications, 20 dietary items, 4 medical procedures, and 15 other factors showed a statistically significant association (FDR < 0.05) with the

microbiome composition, either with microbial diversity or differential abundance on the species-level or functional-level profiles (Supplementary Table 6, Supplementary Fig. 3).

**Associations with long-term usage of medications**. To show the value of continuous linkages with EHRs, we studied the associations between the microbiome and the long-term history of antibiotic (ATC J01) or antidepressant (ATC N06A) usage. We summarized the number of prescriptions filled by the participants in the last 10 years before the sample collection. The subjects were then divided into five distinct classes: the nonusers (subjects not having used any medications within 10 years) and the rest based on the quartiles of the number of prescriptions filled over the 10-year period. To avoid the direct effect of the medication on the microbiome due to recent usage, we excluded the participants who had taken the medication during the last 6 months before sample collection.

There was a significant decrease in the number of observed species with increasing amounts of antibiotics used; however, we observed no major difference in the Shannon's index for either antibiotic or antidepressant usage (Fig. 4a). This is likely due to the exclusion of subjects taking these medications within six months of sample collection. On the contrary, a history of antibiotic or antidepressant usage was associated with the microbiome composition ($R^2 = 0.31\%$ and $R^2 = 0.08\%$, respectively, Fig. 3, Supplementary Table 4). This was characterized by a shift in the first two principal components for both antidepressants and antibiotics (Fig. 4). The association was especially apparent for antibiotics, as higher amounts of antibiotics used over the last 10 years showed changes in PC1 and PC2 towards the *Bacteroides*-dominant part of the PCA plot depicted in Fig. 2c. It is likely that numerous courses of antibiotics over the years are a sign that a person suffers from a health complication but taking numerous courses of antibiotics also leads to changes in the gut microbiome. In our study we observed that as many as three or more courses of antibiotics over 10 years already lead to a shift in the microbial composition in the gut.

**The role of antibiotic usage in common dysbiosis**. Amongst the identified associations between species and diseases (Supplementary Table 5), many were overlapping between different diseases (Fig. 5a). Remarkably, diseases with diverse pathophysiologies like anxiety disorder, hypertension, and gout display a significant number of overlapping associations. Altogether, 109 species showed associations with at least 3 diseases, supporting the previously reported idea of a common dysbiosis between diseases (Supplementary Table 7)[7,15]. Furthermore, species co-occurrence networks showing that the species being associated with diseases for which the overlap in associations is not necessarily high display direct interactions in the co-occurrence networks (Supplementary Fig. 4). For example, Crohn's disease (ICD code: K50) and essential hypertension (ICD10 code: I10) have 30 overlapping associations; however, the networks of these disease-associated species are highly similar due to species co-occurrence (Supplementary Fig. 4).

Based on the previously described effect of long-term antibiotic usage on the microbiome composition, we further analyzed the associations between species and diseases by taking into account the amount of antibiotic usage over the last 10 years as a confounder (Supplementary Table 5). Remarkably, adjusting for long-term antibiotic usage significantly reduced the number of identified associations (Fig. 5b). For example, most of the previously identified associations were lost for gastroesophageal reflux disease (before adjustment 38, after 1), irritable bowel syndrome (before adjustment 16, after 1), and other anxiety

disorders (before adjustment 48, after 3; Fig. 5). Similarly, the overlap between different diseases in terms of the number of associated species was significantly changed, and the number of species having associations with at least 3 diseases was 41 compared to the 109 species found in the previous analysis not adjusting for antibiotic usage. This finding suggests that adjusting for a history of antibiotic usage can largely affect the results of the analysis, even for diseases like anxiety disorder and hypertension, for which antibiotics are not used as a primary treatment.

Due to the increasing interest and potential of using machine learning methods for predicting complex disease based on the microbiome, we also evaluated the predictive value of the microbiome compared to a history of antibiotic usage (Methods; Fig. 5c). Comparison of the average area under the curve (AUROC) values of the elastic net models shows that using information regarding long-term antibiotic usage in addition to conventional predictors (e.g., age, gender, BMI, and stool consistency) has an equal or better predictive power compared to the combination of using microbial features and conventional predictors for almost all of the phenotypes studied (Fig. 5c and Supplementary Table 8). T2D is the only exception, for which the microbial predictors in addition to conventional predictors led to models that had remarkably higher AUROC values compared to the models using a history of antibiotic usage and conventional predictors. Microbial predictors also improved the AUROC for anxiety disorder relative to the *null* model, but a history of antibiotic usage showed even higher AUROC values.

Although we observed a major impact of antibiotic usage on the identified associations, host-targeted medications used for the treatment of complex diseases can also alter the gut microbiome[16]. For example, the antidiabetic drug metformin is known to alter the microbial composition, which makes it challenging to differentiate drug-mediated and disease-mediated effects[17]. Because the adjustment for antibiotic usage did not affect the number of T2D-related associations, we next adjusted the analysis for metformin usage to distinguish between T2D disease and drug effects. After adjusting for metformin usage, 15 species were identified to be associated with T2D, compared with 18 species without adjustment (Supplementary Table 5). Therefore, the usage of host-targeted medications in addition to antibiotics can confound the analysis, making it difficult to identify disease-specific associations.

## Discussion

The gut microbiome is highly individual, and our understanding of the factors associated with the variation in the microbiome composition has certainly improved with large population-based cohort studies[6,7]. These studies make an important contribution with respect to the perspective of personalized medicine, where, in addition to a person's genetic profile and lifestyle, microbial data are included to estimate disease risks and to identify individual-specific drugs for effective treatment. In our study, we analyzed the microbiomes of 2509 individuals from the volunteer-based EstBB. Moreover, we examined the extent to which various factors are associated with the composition of the gut microbiome and identified a number of microbes and pathways showing significant associations with different diseases, medications, as well as intrinsic and lifestyle factors. We provide evidence that significant changes in the microbiome composition are associated with a history of antibiotic usage over a 10-year period, demonstrating the value of continuous linkage with EHRs. Finally, our data supports previous studies that multiple diseases share a common dysbiosis, but this could be partly explained by a history of antibiotic usage.

The gut microbiome profile in the Estonian population is comparable to those reported in previous large-scale studies of

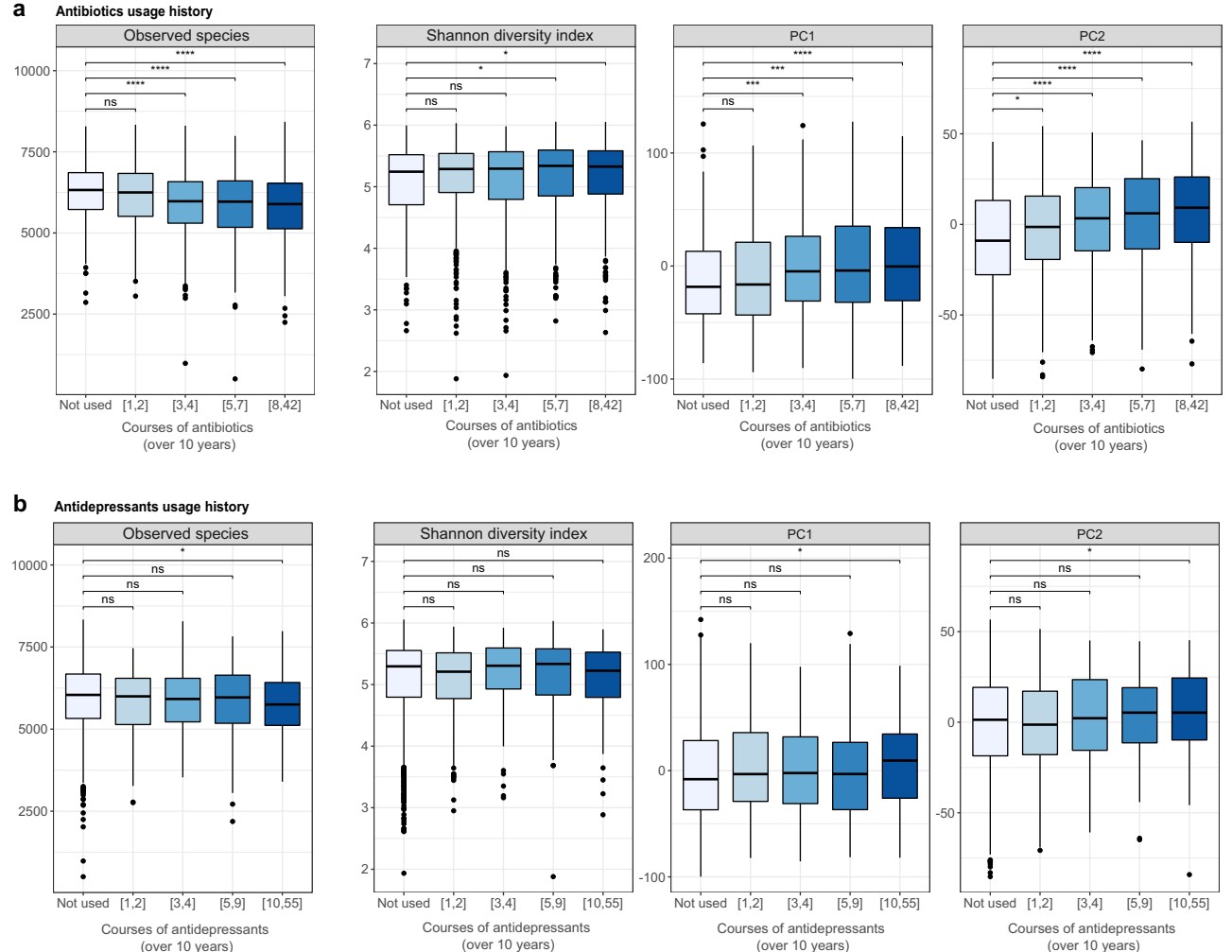

**Fig. 4 Associations of medication usage history with the observed number of species (the y-axis represents the number of species), Shannon diversity (the y-axis represents the Shannon's diversity index), or the first two principal components (PCs) of the species-level microbial composition.** **a** Antibiotics, **b** Antidepressants. Asterisks indicate statistically significant differences between the drug usage history groups using Wilcoxon test (FDR < 0.05*, FDR < 0.01**, FDR < 0.001***, FDR < 0.0001****), and *ns* notes statistically nonsignificant results. Color key indicates the five distinct classes of medication users, the non-users and four additional classes based on the quartiles of the number of prescriptions filled over the 10-year period. The sample size for antibiotics were following: nonusers $n = 243$; [1,2] $n = 549$; [3,4] $n = 440$; [5,7] $n = 395$; [8,42] $n = 400$ and for antidepressants: nonusers $n = 1761$; [1,2] $n = 188$; [3,4] $n = 96$; [5,9] $n = 109$; [10,55] $n = 115$. For all boxplots, the central line, box and whiskers represent the median, interquartile range (IQR), and 1.5 times the IQR, respectively.

European populations, where the same bacterial phyla with similar relative abundances are represented in the gut microbiome[6–8]. Similarly, the functional profile of the gut microbiome in the population is more stable compared to the relative abundance of microbial phyla described previously[7,18,19]. Although the core microbiome depends largely on the analysis methodology and definition[20], we identified a large number of similar genera among European populations[8,11].

In this study, we used extensive clinical and lifestyle data collected from EHRs and questionnaires. While questionnaires provide information regarding a person's lifestyle and diet, EHRs allow access to medically confirmed data on the participants' current as well as incident diseases, medication usage, and medical procedures. Overall, we measured 252 different factors from a broad set of categories that explained 10.14% of the total variability in the species-level microbial composition and 10.50% of the functional variability between individuals. This explained level of variation is comparable with previous studies in European populations and confirms the high individuality of the

microbiome[7]. Furthermore, we saw overlapping results with previous studies in which a lower biodiversity was associated with a loose stool (based on the Bristol scale score), smoking, a higher BMI, and the consumption of soft drinks and preprocessed meat products; while physical activity and the consumption of berries, fruits, and vegetables were related to a higher diversity[6,21]. New significant associations, among others, include medical procedures such as the removal of the cecum and recent anesthesia as well as dietary information like the origin of the drinking water and medications, e.g., glucocorticoids and long-term antibiotic usage.

The possibility of using data from EHRs is a major advantage of the EstMB cohort. We visualized how self-reported questionnaires do not capture the health profiles of the biobank participants comprehensively and observed that the resulting prevalence of diseases and other conditions differs greatly when compared with the data from the EHRs documented by medical specialists. For example, information on vitamin D deficiency comes largely from EHRs, and only a small portion of questionnaires included this information. Another benefit presented

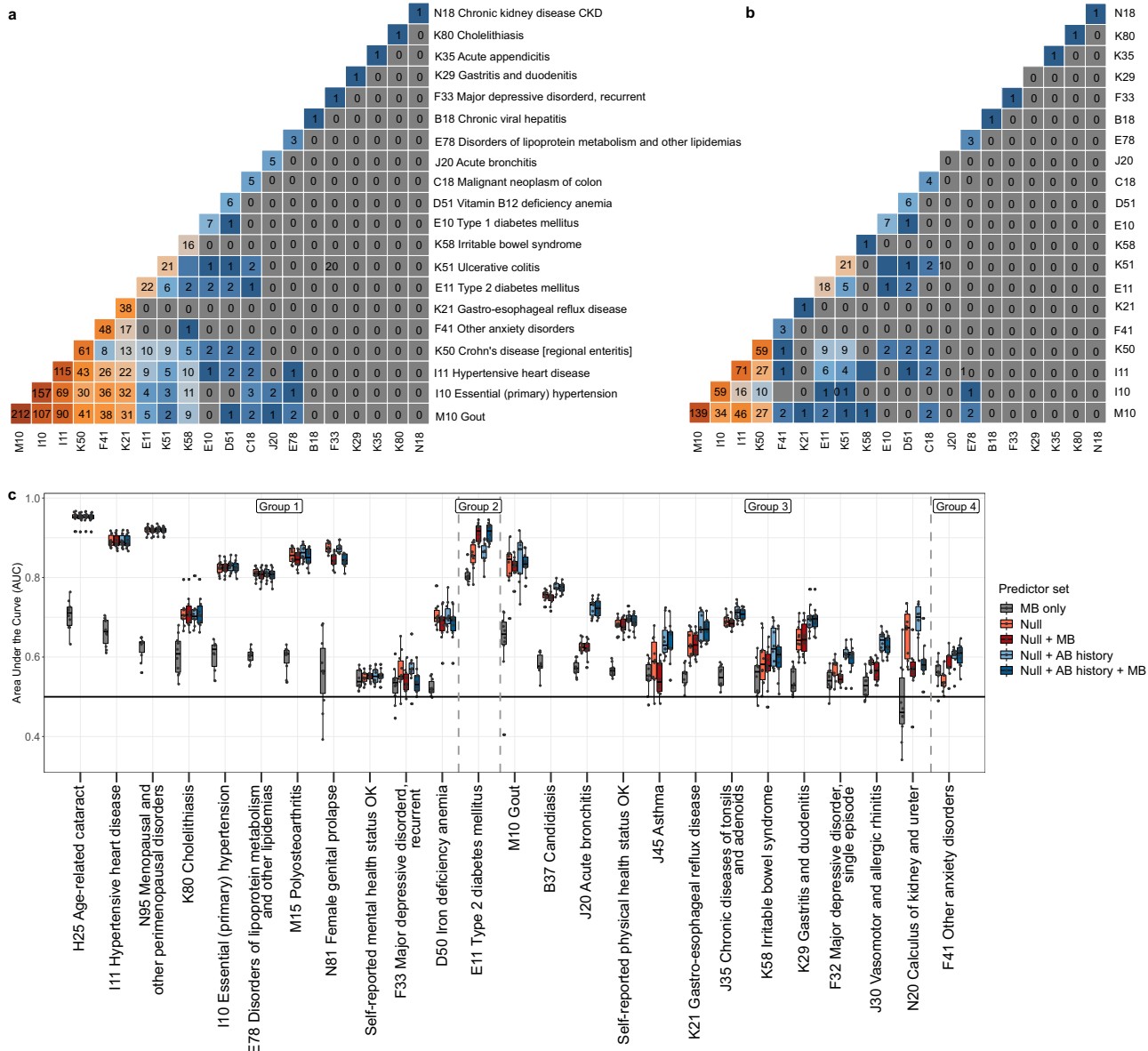

**Fig. 5 Effect of adjusting for antibiotic usage on the number of overlapping associations between various diseases. a** Heatmap of overlapping associations between various complex diseases before adjusting for antibiotic usage. **b** Heatmap of overlapping associations between various complex diseases after taking long-term antibiotic usage into account. **c** The area under the receiver operating curve (AUROC) values on 10 random test sets for elastic net regression models for predicting diseases based on different predictor sets. The *null* model includes age, gender, BMI, and stool consistency as predictors, microbiome (MB), and history of antibiotic (AB) usage. Group 1 consists of phenotypes for which the *null* model provides the best AUROC. Group 2 includes type 2 diabetes, for which the sets including microbial predictors provide the best AUROC. Group 3 includes phenotypes for which the microbiome does not provide an additional predictive value compared to the *null* model and a history of antibiotic usage leads to the best average AUROC. Group 4 includes anxiety disorder, for which the microbial predictors lead to a higher AUROC compared to the *null* model, but a history of antibiotic usage leads to the highest average AUROC. Abbreviations in the x-axis of **a**, **b** and **c** are the international classification of diseases-10 codes (ICD-10) for diagnoses. The precise sample size and ICD- 10 descriptions can be located in the Supplementary Table 5. For all boxplots in **c**, the central line, box and whiskers represent the median, interquartile range (IQR), and 1.5 times the IQR, respectively.

by EHRs is the possibility of having access to information on medical procedures. This allowed us to identify novel significant associations and showed that medical procedures such as tonsillectomy and anesthesia have a significant effect on the microbial composition.

In addition to analyzing the associations with current medication usage, we assessed the effects of long-term usage of antibiotics and antidepressants (10-year period) on the gut microbiome. We observed significant changes in the composition of the microbiome after the participants had taken only 3–4

courses of antibiotics. The fact that a shift in the microbial composition is evident with only a few courses of treatment is intriguing, as half of the participants take more than four courses and Estonians are among the lowest consumers of antibiotics in Europe, suggesting an even stronger effect in other populations[22]. The effects of antibiotic overuse on the normal microbial community structure and health have been reported in both humans and mice[23–26]. It has been shown that after antibiotic treatment, some members of the microbial community do not recover to pretreatment levels and disappear from the community

indefinitely[27–29]. A similar observation, although not so distinct, was visible with antidepressant usage. Moreover, when we took the history of antibiotic usage into account in identifying disease-related microbes, a vast number of associations were lost. Nevertheless, for some diseases such as T2D, Crohn's disease, and ulcerative colitis, the long-term usage of antibiotics did not have a significant effect on the number of detected associations, suggesting a more disease-specific dysbiosis.

Our data also supports the common dysbiosis hypothesis[7,15]. When comparing species–disease associations, we identified common signals for multiple diseases. However, we again saw a profound effect of a history of antibiotic usage. Adjusting for antibiotic usage significantly reduced the number of shared species between the diseases, suggesting that a history of antibiotic usage can at least partly explain the common dysbiosis. In conclusion, long-term antibiotic usage has a significant effect on the overall microbial composition and is an important factor to consider when interpreting the results of association analyses. Disentangling the effects of host-targeted medications and disease is a further complication for identifying disease-specific microbial signals[30]. To illustrate this, we showed that adjusting for metformin usage could help to further disentangle T2D-specific associations. Using our data, the confounding effect of other host-targeted medications on complex diseases could be further investigated in the future.

Obtaining health data through linkage to the EHRs can provide a more reliable diagnosis of diseases, detailed information on medication usage, and information on medical procedures. However, the EHR data also opens up the possibility of investigating adverse drug reactions by studying subjects who rapidly change their medication or examining the effect of a history of long-term medication usage on the microbiome composition as we have shown in this study with antibiotics. In addition, the EHRs include epicrises, that enables us to gain a detailed look into the medical history of an individual, thus allowing us to further refine the analysis by examining disease subtypes and possible comorbidities. Together with the opportunity to call back participants for follow-up examinations, the EHRs provide an opportunity for more in-depth analysis and discovery of novel links between the microbiome and health in the future.

Limitations of our study should be noted when interpreting the results. As the EstMB cohort is volunteer-based, it might not provide a comprehensive overview of the whole Estonian population. Our cohort suffers from gender unbalance—there are more females than males. Also, the study might attract people who are more interested in their health. Furthermore, for some of the medications and diseases, a relatively low number of users and cases were available, which could mean a low statistical power for detecting an effect with the microbiome. Therefore, the associations identified in the current study should be carefully interpreted and require further investigation using larger sample sizes. Lastly, reproducibility in microbiome studies remains challenging due to the complexity of sample collection, data analyses, and different bioinformatic approaches used from study to study[11,31–35].

The EstMB cohort is an outstanding data resource that will help to confirm existing knowledge and provide novel insights about the microbiome, its associated factors, and its potential for identifying prognostic markers. Moreover, analysis of this information will help to raise new questions to be addressed in microbiome studies. Therefore, there is an ever-growing necessity to improve the data resource. Today, the EstMB already includes the possibility of studying the oral microbiome and performing metabolic profiling analysis using plasma or stool samples collected from the EstMB cohort. Incorporating multi-site microbiome data with human genetics and metabolomics can help us to further uncover the pathophysiology of complex diseases and improve decision-making in clinical applications.

## Methods

**EstMB and metadata collection**. The Estonian microbiome cohort was established in 2017, when stool, oral, and blood samples were collected from 2509 EstBB participants (1764 females and 745 males), aged 23–89 years. Participation in the study was voluntary and no compensation was paid. The EstBB is a volunteer-based population cohort initiated in 1999[9] that currently includes over 202,282 genotyped adults (≥ 18 years old) across Estonia. All individuals from the EstBB cohort have been genotyped, and multiple additional datasets are available for a subset of the cohort, including whole-genome sequencing, whole-exome sequencing, and metabolite data using a nuclear magnetic resonance spectroscopy platform (Supplementary Fig. 1). All participants included in the EstMB cohort provided informed consent for the data and samples to be used for scientific purposes. This study was approved by the Research Ethics Committee of the University of Tartu (approval No. 266/T10) and by the Estonian Committee on Bioethics and Human Research (Estonian Ministry of Social Affairs; approval No. 1.1-12/17). All participants have joined the Estonian Biobank on a voluntary basis and have signed a broad consent form, which allows to receive participant's personal and health data from national registries and databases. Rights of gene donors are regulated by Human Genes Research Act (HGRA) § 9 – Voluntary nature of gene donation (https://www.riigiteataja.ee/en/eli/ee/531102013003/consolide/current).

Extensive information was collected on the EstMB participants, including data from both questionnaires (self-reported) and EHRs (completed by medical professionals). The EHR data on the diseases, medications, medical procedures, and causes of death were obtained from the Estonian Health Insurance Fund, Estonian National Death Registry, Estonian Cancer Registry, Estonian Causes of Death Registry, and the two largest hospitals in Estonia (University of Tartu Clinic and North Estonia Medical Centre). EHRs include the data since 2004. We focused on the diseases, medications, and medical procedures recorded in the EHRs to achieve high reliability for the phenotypes of interest, since these are written by medical professionals. The reliability of the diagnosis data from EHRs was further increased by using at least two data entries from the databases to account for wrong entries or misdiagnoses. Arbitrarily selected diseases (based on 3-digit ICD10 categories) with at least 10 cases were chosen for downstream analysis, resulting in 71 diseases (Supplementary Table 1). Although 537 diseases had at least 10 cases, we decided to focus on chronic illnesses (Supplementary Table 9). Medications were grouped into categories based on the ATC classification at the lowest ATC level (up to 7-digit code; ATC level 5) similarly to Gacesa et al.[7] ATC categories with fewer than 10 cases were grouped into a higher level. ATC categories with fewer than 10 cases at any ATC level were removed from the analysis. In total, 132 medications or medication groups were analyzed, of which 90 were classified as ATC level 5 (7-digit code), 26 were classified as ATC level 4 (5-digit code), and 16 were classified as ATC level 3 (4-digit code) (Supplementary Table 1). Also, arbitrary selection of the medical procedures that had been conducted six months prior to the sample collection and included more than 10 cases were studied (Supplementary Table 1). In addition to the EHR data, the patients reported their diseases, medications, medical procedures, and health-related behavior in terms of lifestyle using a microbiome study-specific questionnaire, which included questions about their diet (e.g., dietary frequency questionnaire), physical activity, medical data, living environment, delivery mode, and stool characteristics (Bristol stool scale). The list of all studied variables is listed in Supplementary Table 1.

In addition to the questionnaire and EHR data, the participants' anthropometric measurements (height, weight, blood pressure, and waist and hip circumferences) were taken during a pre-registered visit upon delivering the stool sample. Furthermore, the participants donated an oral buccal swab sample and a blood sample during the visit for further analysis.

**Microbiome sample collection and DNA extraction**. The sample collection took place between 2017 and 2019. The participants collected a fresh stool sample immediately after defecation with a sterile Pasteur pipette and placed it inside a polypropylene conical 15 mL tube. The participants were instructed to time their sample collection as close as possible to the visiting time in the study centre and keep the samples in the fridge (+ 4 °C) until transportation. The participants delivered the sample to the study center, where it was stored at −80 °C until DNA extraction. The median time between sampling and arrival at the freezer in the core facility was 3 h 25 min (mean 4 h 34 min) and the transport time wasn't significantly associated with alpha (Spearman correlation, p-value 0.949 for observed richness and 0.464 for Shannon index) nor beta diversity (p-value 0.061, R-squared 0.0005). Microbial DNA extraction was performed after all samples were collected using a QIAamp DNA Stool Mini Kit (Qiagen, Germany). For the extraction, approximately 200 mg of stool was used as a starting material for the DNA extraction kit, according to the manufacturer's instructions. DNA was quantified from all samples using a Qubit 2.0 Fluorometer with a dsDNA Assay Kit (Thermo Fisher Scientific). A NEBNext® Ultra™ DNA Library Prep Kit for Illumina (NEB, USA) was used for generating sequencing libraries, according to the manufacturer's recommendations. Briefly, 1 μg of DNA per sample was used as the input material.

Index codes were added to attribute the sequences to each sample. The DNA sample was fragmented by sonication to an average size of 350 bp, and then the DNA fragments were end-polished, A-tailed, and ligated with the full-length adaptor for Illumina sequencing with further amplification by the polymerase chain reaction (PCR). Finally, the PCR products were purified (AMPure XP system), and libraries were analyzed for size distribution by an Agilent 2100 Bioanalyzer and quantified using real-time PCR.

**Metagenomics data analyses**. The shotgun metagenomic paired-end sequencing was performed by Novogene Bioinformatics Technology Co., Ltd. using the Illumina NovaSeq6000 platform, resulting in $4.62 \pm 0.44$ Gb of data per sample (insert size, 350 bp; read length, $2 \times 250$ bp). A total of 2514 samples belonging to 2509 individuals were sequenced, including 5 biological replicates from one individual. We observed that the microbiome composition between biological replicates was more similar than between random pairs (Supplementary Fig. 5). First, the reads were trimmed for quality and adapter sequences. The host reads that aligned to the human genome were removed using SOAP2.21 (parameters: -s 135 -l 30 -v 7 -m 200 -x 400)[36]. Quality controlled data of each sample was then used for metagenomic assembly using SOAPdenovo (v. 2.04, parameters: -d 1 -M 3 -R -u –F)[37]. Next, SOAP2.21 was used to map the clean data of each sample to the assembled scaftigs (i.e., continuous sequences within scaffolds). Unutilized paired-end reads of each sample were put together for mixed assembly. MetaGeneMark (v.3.38) was used to carry out gene prediction (gene length > 100 bp) based on the scaftigs ($\geq$ 500 bp), which were assembled by single and mixed samples. CD-HIT (v.4.6) was used to dereplicate the predicted genes based on 95% identity and 90% coverage to generate the gene catalogues (parameters: -c 0.95, -G 0, -aS 0.9, -g 1, -d 0)[38]. The longest dereplicated gene was defined as the representative gene (i.e., unigene). SoapAligner[39] (v.2.21, parameters: -m 200, -x 400, identity $\geq$ 95%) was then used to map the clean data to the gene catalogues and to calculate the quantity of the genes for each sample. The gene abundance was calculated based on the total number of mapped reads and the normalized gene length. The taxonomic composition of the metagenomes was identified by comparing the marker gene homologs to a NCBI nonredundant NCBI-nr (ftp://ftp.ncbi.nlm.nih.gov/blast/db/) database (201810) of taxonomically informative gene families using DIAMOND (v0.9.9.110)[40]. The homologs were annotated based on the sequence or phylogenetic similarity to the database sequences. The abundance of different taxonomic ranks was based on gene abundance tables. Microbial functional pathways were annotated using the KEGG database (https://www.genome.jp/kegg/).

**Filtering microbiome data**. For downstream analysis, we removed three individuals with an exceptionally low number of reads. In total, 17,158 species and 7869 KOs were identified. Alpha and beta diversity analyses were carried out on the whole identified composition. To identify univariate associations with species and KOs, species and KOs that were detected with > 10% prevalence at a relative abundance of 0.1% were used in order to limit the number of tests carried out, resulting in 1231 species and 1974 KOs. Similarly, filtered species were used as predictors for building classification models. We did not rarefy the counts to avoid loss of data.

**Statistics and reproducibility**. The EstMB cohort is a volunteer-based population cohort. No statistical method was used to predetermine the sample size, sample collection was not randomized and no exclusion criteria were applied. The statistical analysis workflow used in our study is based on standard statistical techniques applied in microbiome studies. All statistical analyses were carried out using R (v. 4.0.1) software.

**Microbiome diversity analysis**. The observed number of unique species and the Shannon diversity index were used to assess the alpha diversity using the vegan package (v2.5.6). The observed number of unique KOs was used to characterize the diversity of the functional profile. To associate the alpha-diversity metrics with the phenotypes of interest, the Spearman correlation coefficient was used. The Euclidean distance on the CLR-transformed microbiome (species and KO) profile was used to calculate the between-sample distances for the beta diversity analysis. Permutational analysis of variance (PERMANOVA) on the Euclidean distances was used to test the associations between the phenotypes and microbiome composition using 10,000 permutations for the p-value calculations. PERMANOVA was carried out using the adonis function in the vegan package (v.2.5-6.). To apply the CLR transformation, zero counts were imputed with a pseudo count of 0.5. The Benjamini–Hochberg procedure was used to account for multiple testing.

**Core microbiome detection**. Genera with a prevalence of more than 10% using a detection threshold of 0.1% were considered core genera.

**Detecting microbial features associated with the phenotypes**. The ANOVA-like Differential Expression tool (ALDEx2, v.1.18.0) was used to identify species and KOs associated with prevalent diseases, medications, and dietary factors. Models were adjusted for gender, age, BMI, and stool consistency. The number of antibiotic prescriptions over the last 10 years before sample collection was used as an additional covariate to assess the role of antibiotic usage history on the detection

of disease-specific associations. Metformin usage was used as an additional covariate for disentangling the T2D-metformin effects. Subjects who had taken antibiotics within the last 6 months of sample collection were excluded from the analysis ($n = 482$). Further, participants who had missing values for gender, age, BMI or stool consistency, had their samples removed ($n = 73$), resulting in 1951 samples for the analysis. The number of cases for diseases and medication users after excluding antibiotic users and subjects with missing values in covariates is summarized in Supplementary Table 5. Zero imputation, number of Monte-Carlo instances, and selection of the denominator followed the default behavior of the aldex.clr function.

**Prediction models for common diseases**. Elastic net regression was used to build models for predicting disease prevalence using different sets of predictor variables. As a *null* model, age, gender, BMI, and stool consistency were used as predictors. The *null + AB* model included the number of antibiotics used within the last 10 years as a predictor in addition to the predictors of the *null* model to assess the added predictive value of antibiotic usage. The *null + MB* model included CLR-transformed abundances of 1231 species in addition to the predictors of the *null* model to assess the added predictive value of the microbiome relative to the *null* model. The *null + MB + AB* model included antibiotic usage and CLR-abundances of species in addition to the predictors of the *null* model to assess whether the microbiome can provide additional predictive value compared to the *null + AB* model.

Elastic net models were implemented in R using the tidymodels (v0.1.1) and glmnet (v3.0-2) packages. To build the models, data were split in a 75:25 ratio to the training and test datasets, and models were tuned on the training data using a 5-fold cross-validation and grid search with 100 hyperparameter combinations. The initial data split and cross-validation splits were stratified by the disease occurrence to deal with class imbalance. The performance of the models was evaluated on the test dataset using the AUROC. The model building and evaluation were repeated 10 times on random training-test splits to gain a robust measure of performance. The average AUROC of the 10 repetitions was reported. Diseases showing association with the beta-diversity with at least 50 cases after excluding antibiotic users and subjects with missing values for gender, age, BMI or stool consistency were considered (Supplementary Table 5).

**Detection of species co-occurrence**. Species co-occurrence networks were calculated using SPIEC-EASI (v1.1.0)[41] with default parameters and the (bounded) StARS model selection. Isolated nodes were excluded from the network visualization.

**Reporting summary**. Further information on research design is available in the Nature Research Reporting Summary linked to this article.

## Data availability

The metagenomic data generated in this study have been deposited in the European Genome-Phenome Archive database (https://www.ebi.ac.uk/ega/) under accession code EGAS00001008448. The phenotype data contain sensitive information from healthcare registers and they are available under restricted access through the Estonian biobank upon submission of a research plan and signing a data transfer agreement. All data access to the Estonian Biobank must follow the informed consent regulations of the Estonian Committee on Bioethics and Human Research, which are clearly described in the Data Access section at https://genomics.ut.ee/en/content/estonian-biobank. A preliminary request for raw metagenome and phenotype data must first be submitted via the email address releases@ut.ee. Used databases are NCBI nonredundant (NCBI nr) database 201810 (https://www.ncbi.nlm.nih.gov/blast/db/) and KEGG (https://www.kegg.jp/).

## Code availability

The source code for the analyses is available at https://doi.org/10.5281/zenodo.5767071.

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

## Acknowledgements

The authors would like to thank Mari-Liis Tammeorg, Marili Palover, Anu Reigo, Neeme Tõnisson, Liis Leitsalu, and Esta Pintsaar for participating in the sample collection process. We thank Steven Smit, Rita Kreevan, Martin Tootsi, and Triinu Temberg for the DNA extraction process. We also thank all of the study participants. This work was funded by an Estonian Research Council grant (No. PUT 1371 to EO) and an EMBO Installation grant (No. 3573 to EO), by the European Regional Development Fund (Project No. 15-0012 GENTRANSMED to AM) and the Estonian Center of Genomics/Roadmap II (project No. 16-0125 to AM). AM was supported by Estonian Research Council grant PUT (No. PRG 687). OA, KL, and KLK were supported by The European Regional Development Fund (Smart specialization PhD scholarships).

## Author contributions

E.O. designed and supervised the study. K.L.K., K.L., E.O., and A.M. organized the collection of the samples. K.L.K. organized the phenotype and health data from questionnaires and electronic health records. O.A. performed the data analysis. O.A. and K.L.K. interpreted the data and prepared the figures. O.A., K.L.K., K.L., and E.O. wrote the paper. All authors read and approved the final paper. These authors contributed equally: O.A., K.L.K.

## Competing interests

The authors declare no competing interests.
