## [Peer Review File · Nature Communications]

REVIEWER COMMENTS

Reviewer #1 (Remarks to the Author):

Review Aasmets et al – comments to authors

Aasmets and co-workers present a well-executed analysis of intestinal microbiota variation within an Estonian population cohort. The study follows generally accepted practices as well as analysis approaches. My main worry however is that, although I can much appreciate the effort and size of the study, as well as the generation of a nice dataset for the research community, the large majority of the biological findings presented are already known: taxonomic variation vs functional stability, dominance of Bacteroidetes & Firmicutes, the major microbiota covariates (BMI-associated, transit-time associated, drugs (incl antibiotics and antidepressants)). The only area that is deepened a bit is that antibiotics-associated associations confound disease prediction/disease-microbiota associations. Although this topic is expanded here also into long-term usage, its not that this is something that is completely novel either. The shotgun data is underused – very little functional insights are given, strain-level analysis is not done. Data is not quantitative, no integration with other omics data is provided, no novel analysis approach is used, no experimental follow-ups are provided, etc etc.

From a scientific point of view, I am pleased to see that this study provides dearly needed replication for many of these previous studies. However, the lack of new insights unfortunately undermines the suitability of this manuscript for a journal like Nature Communications, where novelty and scientific advance is still an important criterion from my perspective.

One additional note: samples were collected fresh and stored at -20 until processing. This is a procedure, which is very risky if the process is not monitored and/or executed properly. How long were samples at room temperature between sampling and freezing? Did this vary between study subjects? Were buffers used and/or anoxic conditions and/or cooling provided for transport? How long were samples at -20 before processing? Were samples extracted in batches or as they came in? How much time difference between the first and last sample?

Reviewer #2 (Remarks to the Author):

The paper for Aasmets et al nicely describes the one extra population-based microbiome study performed on Estonian cohort. The biggest advantage compare to many other population studies is an access to electronic health records – the opportunity barely present in MWAS.

I have no general complaints about the paper and the research itself. There are although few remarks, mostly about phrasing and some extra clarifications to consider.

Minor comments:

1. “study host-related factors, including metagenomic data” I don’t think the “data” is host-related factor. The microbiome by itself is a host-related factor, not the data representing it.
2. During reading, I was struggling with a term ‘procedure’ often used in the paper. (The point that those were ‘medical procedures’ first occurred only in the section ‘overview of microbiome-phenotype associations’, far after the term was used the first time. Can it be replaced to ‘clinical treatment’ or something similar? Otherwise the ‘procedure’ should always be used in text as ‘medical procedure’
3. I guess the statement “Therefore, we focused primarily on the EHRs” should be better clarified. For sure EHRs are more precise than self-reported diseases; however, I guess it makes sense to report some 'benchmarks' of comparison if self-reported data shows substantial concordance with EHRs in a pattern of associations with microbiome traits. I guess it would nicely complement the paper overall and will lead to some conclusions like "self-reported data shows XX% decrease in power compared to EHRs", or "use of EHRs and self-reports, being combined, might increase the power to XX%". Authors have really unique dataset to explore this topic in more detail.
4. In the description of Landscape of Estonian gut microbiome ,the DNA extraction kit should be mentioned as being one of the major drivers of the gut microbiome composition of a studied population.
5. “For example, the beta blockers metoprolol and nebivolol were found to be associated with the gut microbiome, with metoprolol describing more variation in the gut microbiome”. Is this statement takes into account the number of drug users? With higher predictor variance, one can expect higher explained variance in the outcome...
6. “a history of antibiotic or antidepressant usage was associated with variability in the microbiome composition”. I wouldn’t use ‘variability’ here. Just ‘associated with the microbiome composition’
7. “a shift in the values of the first two principal components” -> a shift in the first two principal components
8. “higher values for PC1 and PC2”. Higher/lower values in PCs mean nothing. It’s better to rephrase like ‘show changes in PC1/2 towards...’

9. "Although it is reasonable to assume that numerous antibiotic courses over the years refer to health complications, the shift was already evident for the subjects with more than three antibiotic courses over the last 10 years." – there's no real contradiction between first and the second statement (3 courses still could indicate 'poor health'). Worth to change the phrase to make it more descriptive and less conclusive.

10. Regarding the disease prediction, it makes more sense to perform cross-validation instead of the single training/test split. I will allow to calculate not only a single AUC but also its standard error

Reviewer #3 (Remarks to the Author):

In this study, the authors studied the relationship between the gut microbiome and 252 clinical factors (various diseases, medications, diet, procedures, etc). The data came from 2509 participants in the Estonian Biobank (EstBB), which includes stool, questionnaire, and multi-omic data linked to EHRs. The authors present results describing the variation in microbiome across their cohort and numerous associations with diseases, antibiotic use, and other factors.

This study builds on prior work in this area and provides additional evidence of the importance of the gut microbiome in understanding disease. It is a nice example of the types of analyses that are possible through deep phenotyping that integrates multimodal data. I learned about the EstBB, its Microbiome Cohort, and the rich data it contains. This sounds like a great resource!

Major comments:

1) I don't understand how the 252 clinical features were selected. The authors wrote that the diseases and procedures were "arbitrarily selected". I think the 136 medications are also probably just a subset of what is potentially available from the EHRs. However, some of this seems modeled after their reference #7, which compared the gut microbiome in patients from the Dutch Microbiome Project to 241 clinical factors. If the EHR data are available for these patients, why not use all diagnoses, medications, and procedures, rather than a subset?

2) The issue with selecting arbitrary EHR variables that it is hard to extrapolate the results of this study. For example, the authors found that 75 of the 241 factors (31%) were associated with beta diversity. This sounds impressive; however, there are nearly 70,000 ICD10 diagnosis codes that they did not consider. Did they select these factors to enrich the chances that associations would be

found? As another example, they report that the studied factors describe 10.14% of the microbial species variability. Could this number have been much higher if they used the full EHR data?

Minor comments:

3) This is probably not too relevant to this study, but I found it strange that Figure 1d shows 75% of acute appendicitis cases came from questionnaires only and were not in the EHR data. Do the EHR data only go back 10 years (and miss all the cases the older patients in the cohort had in their youth), while the questionnaires ask for anytime in a patient's life?

Respond to reviewers

December 08, 2021

We thank You for the opportunity to resubmit a revised manuscript “Gut metagenome associations with extensive digital health data in a volunteer-based Estonian microbiome cohort” for publication in Nature Communications. We very much appreciate the effort put into reviewing the manuscript and are grateful for the valuable comments and suggestions brought out by the Reviewers. The constructive suggestions have allowed us to improve the manuscript and state our ideas more clearly. We have addressed the comments of the Reviewers and provide a point-by-point response below.

Reviewer #1 (Remarks to the Author):

Aasmets and co-workers present a well-executed analysis of intestinal microbiota variation within an Estonian population cohort. The study follows generally accepted practices as well as analysis approaches. My main worry however is that, although I can much appreciate the effort and size of the study, as well as the generation of a nice dataset for the research community, the large majority of the biological findings presented are already known: taxonomic variation vs functional stability, dominance of Bacteroidetes & Firmicutes, the major microbiota covariates (BMI-associated, transit-time associated, drugs (incl antibiotics and antidepressants)). The only area that is deepened a bit is that antibiotics-associated associations confound disease prediction/disease-microbiota associations. Although this topic is expanded here also into long-term usage, it's not that this is something that is completely novel either. The shotgun data is underused – very little functional insights are given, strain-level analysis is not done. Data is not quantitative, no integration with other omics data is provided, no novel analysis approach is used, no experimental follow-ups are provided, etc etc.

From a scientific point of view, I am pleased to see that this study provides dearly needed replication for many of these previous studies. However, the lack of new insights unfortunately undermines the suitability of this manuscript for a journal like Nature Communications, where novelty and scientific advance is still an important criterion from my perspective.

We thank the Reviewer for their comments. We agree with the Reviewer's general view that our cohort has great potential to carry out more in-depth data analysis, e.g. to investigate the relationship between health and the microbiome by combining other data layers (multi-omics datasets) and that experimental follow-up studies are needed to demonstrate a causal relationship. These are certainly very important aspects that will definitely be implemented in future projects when focusing on more specific research questions and phenotypes in the EstMB cohort. Our current study characterizes the gut metagenomic profile of an additional population cohort, the Estonian Biobank, where the subjects originate from the entire country. One of the important points of this study was to show the usefulness of long-term electronic health data in microbiome analysis. Access to electronic health data allows the health of the subjects to be monitored retrospectively over a long period of time, thus obtaining more accurate information about a person's health profile. In the present work, we clearly demonstrated the effects of long-term antibiotics usage on gut microbiome and its confounding effect on disease-microbiota associations and predictions. To our current knowledge, the importance of long-term antibiotic usage effects on microbiome associations excluding recent antibiotics usage has not been shown in a large-scale population study. Information about long term-antibiotic usage is relatively easy to obtain/ask from the study subjects and it should be considered when designing future studies. In addition to antibiotics, we showed the effects of long-term use of antidepressants on the gut microbiome. We agree that the reproducibility of biological findings is a critical issue in the microbiome field. We thereby think that confirmation of previously reported associations using EHR data is valuable and its importance for future research can not be undervalued.

One additional note: samples were collected fresh and stored at -20 until processing. This is a procedure, which is very risky if the process is not monitored and/or executed properly. How long were samples at room temperature between sampling and freezing? Did this vary between study subjects? Were buffers used and/or anoxic conditions and/or cooling provided for transport? How long were samples at -20 before processing? Were samples extracted in batches or as they came in? How much time difference between the first and last sample?

We thank the Reviewer for pointing out these important concerns. We further clarified the collection procedures in the manuscript [p. 20, lines 448-458]. The samples were collected within a 2 year time period (2017-2019) covering all seasons (**Review Figure 1**). The participants were instructed to time their sample collection as close as possible to the visiting time in the study center and keep the samples in the fridge (+4C) until transportation. Upon arrival at the research facility, the samples were immediately frozen. Thanks to the reviewer's comment, we noticed an error in our manuscript - the samples were not actually stored at -20C at the research facility, but in -80C until processing. This was now corrected in p. 20. DNA was extracted from the samples after all samples were collected. The median time from sampling to freezing was 3h 25 minutes (average time 4h 34min). The median and average transportation time has now been included in the Methods section [p.20 lines 454-455] (under the "Microbiome sample collection and DNA extraction"). The transport time wasn't significantly associated with alpha (Spearman correlation, p-value 0.949 for observed richness and 0.464 for Shannon index) nor beta diversity (p-value 0.061, R-squared 0.0005). As pointed out by the Reviewer, we replicated findings from several studies that were using different collection methods, which makes us confident about the soundness of our approach for large-scale sample collection.

Review Figure 1. The frequency of the day of the year when the samples were drawn.

Reviewer #2 (Remarks to the Author):

The paper for Aasmets et al nicely describes the one extra population-based microbiome study performed on the Estonian cohort. The biggest advantage compared to many other population studies is an access to electronic health records – the opportunity barely present in MWAS. I have no general complaints about the paper and the research itself. There are although few remarks, mostly about phrasing and some extra clarifications to consider.

We thank the Reviewer for a thorough review! Please find below the answers to the questions and references to the changes made in the manuscript.

Minor comments:

1. "study host-related factors, including metagenomic data" I don't think the "data" is host-related factor. The microbiome by itself is a host-related factor, not the data representing it.

We agree with the Reviewer's comment and apologize for this statement. This has now been corrected to "microbial community" [p. 2, lines 44-45].

2. During reading, I was struggling with a term 'procedure' often used in the paper. The point that those were 'medical procedures' first occurred only in the section 'overview of microbiome-phenotype associations', far after the term was used the first time. Can it be replaced to 'clinical treatment' or something similar? Otherwise the 'procedure' should always be used in text as 'medical procedure'

We kindly accept the suggestion made by the Reviewer. In order to make the article more readable, the term "medical procedure" is now used throughout the article instead of only using the term "procedure".

3. I guess the statement "Therefore, we focused primarily on the EHRs" should be better clarified. For sure EHRs are more precise than self-reported diseases; however, I guess it makes sense to report some 'benchmarks' of comparison if self-reported data shows substantial concordance with EHRs in a pattern of associations with microbiome traits. I guess it would nicely complement the paper overall and will lead to some conclusions like "self-reported data shows XX% decrease in power compared to EHRs", or "use of EHRs and self-reports, being combined, might increase the power to XX%". Authors have really unique dataset to explore this topic in more detail.

We thank the Reviewer for this comment. We agree that such a benchmark would provide further information about the possibilities of using each data collection method. However, such power calculation is highly dependent on the phenotype and the analysis carried out. Review Figure 2 corresponds to the Figure 1d in the manuscript but instead of proportion of all occurrences shows the absolute number of cases. This further highlights that, even if the results obtained by the different data collection methods are concordant, the higher number of cases in EHR for nearly all of the diseases leads to increased power. Also, we emphasize that the level of details in the questionnaire as well as the collection method (answering the questionnaire alone or together with a medical professional) are likely to vary in other cohorts. Thus, in our opinion, the estimates about change in power are highly cohort and study specific and would not lead to a general conclusion. Taken together, we agree that this would be an interesting question to explore, in particular for checking the quality of the questionnaire data, but it requires a thorough and systematic approach across various analyses and phenotypes to reach a general conclusion, which is out of the scope of this manuscript.

Review Figure 2. Number of cases originating from EHRs only, questionnaires only or overlapping both in questionnaires and EHRs.

4. In the description of Landscape of Estonian gut microbiome, the DNA extraction kit should be mentioned as being one of the major drivers of the gut microbiome composition of a studied population.

We thank the Reviewer for this remark. We agree that the DNA extraction kit has a big influence on the microbiome composition and knowledge of the used extraction kit makes the study more reproducible by others. We included the name of the extraction kit in the description of the “Landscape of Estonian gut microbiome”. [p. 6, line 127-129]

5. “For example, the beta blockers metoprolol and nebivolol were found to be associated with the gut microbiome, with metoprolol describing more variation in the gut microbiome”. Is this statement takes into account the number of drug users? With higher predictor variance, one can expect higher explained variance in the outcome...

We thank the Reviewer for pointing out this concern. There is indeed a difference in the number of cases between the users of the two drug groups (Supplementary table 1). However, we did not account for the number of drug users in the analysis similarly to previous studies [Refs: 1, 2] to allow comparability.

6. “a history of antibiotic or antidepressant usage was associated with variability in the microbiome composition”. I wouldn't use ‘variability’ here. Just ‘associated with the microbiome composition’

We agree with this suggestion pointed out by the Reviewer. This has now been corrected. [p. 10, line 235]

7. “a shift in the values of the first two principal components” -> a shift in the first two principal components

We thank the Reviewer for this adjustment. This sentence has now been changed in the manuscript accordingly. [p. 10, lines 237-238]

8. “higher values for PC1 and PC2”. Higher/lower values in PCs mean nothing. It’s better to rephrase like ‘show changes in PC1/2 towards...”

We thank the Reviewer for this suggestion. This has now been changed accordingly. [p 10, line 240]

9. “Although it is reasonable to assume that numerous antibiotic courses over the years refer to health complications, the shift was already evident for the subjects with more than three antibiotic courses over the last 10 years.” – there’s no real contradiction between the first and the second statement (3 courses still could indicate ‘poor health’). Worth to change the phrase to make it more descriptive and less conclusive.

We thank the Reviewer for the comment. In order to make the statement less conclusive, we changed it as follows: “It is likely that numerous courses of antibiotics over the years are a sign that a person suffers from a health complication but taking numerous courses of antibiotics also leads to changes in the gut microbiome. In our study we observed that as many as three or more courses of antibiotics over 10 years already lead to a shift in the microbial composition in the gut” [p.11 line 241-247].

10. Regarding the disease prediction, it makes more sense to perform cross-validation instead of the single training/test split. I will allow to calculate not only a single AUC but also its standard error.

We thank the Reviewer, this is an important comment. In our manuscript, cross-validation was used for identifying the optimal hyperparameters and the performance of the fitted model was evaluated on the independent test set [p. 24, lines 543 - 551]. Independent test set was used to obtain a proper measure of model performance [Refs: 3, 4, 5]. Furthermore, this procedure was repeated 10 times using different initial train-test splits each time. We and others have previously shown how the test set performance can vary depending on the initial data split [Refs: 6, 7, 8, 9]. Thus, multiple data splits helped us to further characterize the generalization performance of the prediction models.

Reviewer #3 (Remarks to the Author):

In this study, the authors studied the relationship between the gut microbiome and 252 clinical factors (various diseases, medications, diet, procedures, etc). The data came from 2509 participants in the Estonian Biobank (EstBB), which includes stool, questionnaire, and multi-omic data linked to EHRs. The authors present results describing the variation in microbiome across their cohort and numerous associations with diseases, antibiotic use, and other factors.

This study builds on prior work in this area and provides additional evidence of the importance of the gut microbiome in understanding disease. It is a nice example of the types of analyses that are possible through deep phenotyping that integrates multimodal data. I learned about the EstBB, its Microbiome Cohort, and the rich data it contains. This sounds like a great resource!

We thank the Reviewer for the kind and supportive review!

Major comments:

1) I don’t understand how the 252 clinical features were selected. The authors wrote that the diseases and procedures were "arbitrarily selected". I think the 136 medications are also probably just a subset of what is potentially available from the EHRs. However, some of this seems modeled after their reference #7, which compared the gut microbiome in patients from the Dutch Microbiome Project to 241 clinical factors. If the EHR data are available for these patients, why not use all diagnoses, medications, and procedures, rather than a subset?

EHRs data are indeed very extensive as they contain records of all visits to the medical centers, medical procedures, diagnoses and prescribed medications. For example, EHR data may also include procedures and diagnoses which are not biologically relevant, such as “Visit to the General practitioner for medical overview”, “Health check for drivers license” etc. Therefore, a selection is needed in order to eliminate such irrelevant factors and also reduce the number of tests performed. Our aim was to obtain as reliable diagnosis, medication usage and medical procedure information from

the EHR as possible. The structure of the Estonian registers and hospital databases together with the possibilities for evaluating the diagnose reliability is described in detail in the Leitsalu et al 2015 study [Ref 10]. Next, we focused on diagnoses, medications and procedures with at least 10 cases. Medications were grouped into categories based on the ATC classification including as many cases in the most precise ATC category as possible. Categories from the most precise (ATC level 5, chemical substance level; 7-digit code) up to more general (ATC level 3; pharmacological subgroup; 4 digit code) were used (Review Figure 3). When the more precise ATC level had less than 10 cases, they were grouped into a more general level (described in Review Figure 3). This allowed us to analyze the usage of as many different medications or medication groups as possible. In total, we identified 1142 unique ICD10 codes for EstMB cohort individuals (Review Figure 4). Out of the 1142 ICD10 codes, 537 ICD codes had at least 10 cases. Many ICD10 codes out of the 537 are not meant to describe the health status of a person, thus they should be excluded from the analyses. For example, 3 most abundant ICD10 codes are “Encounter for medical observation for suspected diseases and conditions ruled out (Z03)”, “Encounter for other special examination without complaint, suspected or reported diagnosis (Z01)”, and “Encounter for administrative examination (Z02)”. Therefore, the selection of ICD10 codes is important. We focused primarily on chronic conditions as they can be considered prevalent during the stool collection independent of the time of the first diagnosis. For example, “Acute upper respiratory infections (J06)” being one of most abundant ICD codes with 1270 cases can potentially influence microbiome, but only recent cases before the sample collection should likely be studied.

Review Figure 3. Grouping process for medication ATC levels.

Review Figure 4. Number of ICD10 codes by the number of cases in EstMB.

2) The issue with selecting arbitrary EHR variables that it is hard to extrapolate the results of this study. For example, the authors found that 75 of the 241 factors (31%) were associated with beta diversity. This sounds impressive; however, there are nearly 70,000 ICD10 diagnosis codes that they did not consider. Did they select these factors to enrich the chances that associations would be found? As another example, they report that the studied factors describe 10.14% of the microbial species variability. Could this number have been much higher if they used the full EHR data?

As addressed in our previous comment, the EHR data includes a vast amount of data together with less relevant information about one's health and selection of the ICD10 codes needs to be done. We agree that the number of seen associations depends on the variables chosen for the analysis and the percentage of variability described can be higher if more factors were included and this percentage is relative to the chosen factors. However, most of the diseases used in the analysis are amongst the most studied in the microbiome field. With previous results and described associations available, the selection of the diseases is definitely justified. We emphasize that the selection of the factors was done before the analysis. We have now supplemented the manuscript with data showing the number of cases for all of the 1142 ICD10 codes detected [p. 19, lines 427-428] (Supplementary Table 9). We agree that the described percentage of the microbial species level variability can likely be improved with the EHR records, but this was not the primary aim of the study. We thank the reviewer once again for the thought and hope to answer this question in the future.

Minor comments:

3) This is probably not too relevant to this study, but I found it strange that Figure 1d shows 75% of acute appendicitis cases came from questionnaires only and were not in the EHR data. Do the EHR data only go back 10 years (and miss all the cases the older patients in the cohort had in their youth), while the questionnaires ask for anytime in a patient's life?

The questionnaires indeed ask for data about any time in the patient's life, while EHRs include the data from 2004 when the national digital health systems were established. This information was included in the methods section [p. 19, line 425]. Diseases that participants had in their childhood are not always depicted in neither questionnaire nor EHRs as patients either do not remember them or do not have any complications-medications related to the disease which would make them appear on EHRs in a later stage, as would be the case with chronic diseases. Therefore, indeed some diagnosis might be missed by EHRs or both questionnaires and EHRs.

Sincerely yours,

Elin Org, PhD

Associate Professor

Estonian Genome Centre, Institute of Genomics

University of Tartu

References

- [1] "Population-based metagenomics analysis reveals markers for gut microbiome composition and diversity" ([10.1126/science.aad3369](https://doi.org/10.1126/science.aad3369))
- [2] "Population-level analysis of gut microbiome variation" ([10.1126/science.aad3503](https://doi.org/10.1126/science.aad3503))
- [3] "Stool Studies Don't Pass the Sniff Test: A Systematic Review of Human Gut Microbiome Research Suggests Widespread Misuse of Machine Learning" ([arXiv:2107.03611](https://arxiv.org/abs/2107.03611))
- [4] "Test set verification is an essential step in model building" (<https://doi.org/10.1111/2041-210X.13495>)
- [5] "Navigating the pitfalls of applying machine learning in genomics" (<https://doi.org/10.1038/s41576-021-00434-9>)
- [6] "Learning sparse log-ratios for high-throughput sequencing data" (<https://doi.org/10.1093/bioinformatics/btab645>)
- [7] "Tree-aggregated predictive modeling of microbiome data" (<https://doi.org/10.1038/s41598-021-93645-3>)
- [8] "A Framework for Effective Application of Machine Learning to Microbiome-Based Classification Problems" ([10.1128/mBio.00434-20](https://doi.org/10.1128/mBio.00434-20))
- [9] "Machine Learning Reveals Time-Varying Microbial Predictors with Complex Effects on Glucose Regulation" (<https://doi.org/10.1128/mSystems.01191-20>)
- [10] Cohort Profile: Estonian Biobank of the Estonian Genome Center, University of Tartu. (<https://doi.org/10.1093/ije/dyt268>)

REVIEWERS' COMMENTS

Reviewer #2 (Remarks to the Author):

All my concerns are addressed, so I don't see any other complaints that prevents this paper from being published. I would again highlight the importance and novelty of EHR use in the manuscript: this is really an important advantage and it indeed makes the paper strong enough for the journal

Reviewer #3 (Remarks to the Author):

My main comment from before was about the process of selecting the EHR variables. The authors' response was that "a selection is needed in order to eliminate such irrelevant factors and also reduce the number of tests performed". I disagree with this. This study is trying to discover what factors are associated with the metagenome, but they are implying that they already know what is relevant. EHR data can often be associated with outcomes in unexpected ways that do not seem biologically relevant. Even something like "health check for drivers license" tells you that the patient is healthy enough, both physically and mentally, to drive and operate a vehicle--for example, the patient is not in the ICU, is not immobile, and is able to see (the authors include cataract and glaucoma as factors). Reducing the number of tests performed is also not required if the computational resources are available. I would have preferred if this study looked at all EHR variables and showed that the ones that were statistically significant also aligned with the literature. Instead, the authors took diagnoses that they thought were relevant and used this study to confirm which ones really are. They missed the opportunity to discover something unexpected. I still think this is a really interesting paper with important results. I'm just pointing out that there might much more to learn from the amazing database the authors used.

The authors also mention in their response that their "aim was to obtain as reliable diagnoses from the EHR as possible", referencing a 2015 paper by Leitsalu describing the Estonian Biobank. In that paper, Leitsalu defines four levels of "reliability", with some diagnoses being more reliable if, for example, the diagnosis was both confirmed by the doctor and listed in the database. Does this mean that this new study only used the Class 1 reliable diagnoses from the biobank and ignored the other diagnoses? If so, then that should be stated in the manuscript.

Respond to reviewers letter

January 18, 2022

We have addressed the final comments of the Reviewers and provide a point-by-point response below.

REVIEWERS' COMMENTS

Reviewer #2 (Remarks to the Author):

All my concerns are addressed, so I don't see any other complaints that prevents this paper from being published. I would again highlight the importance and novelty of EHR use in the manuscript: this is really an important advantage and it indeed makes the paper strong enough for the journal

We kindly thank the Reviewer for finding that our study makes an important contribution to microbiome research and for providing important comments to improve our manuscript.

Reviewer #3 (Remarks to the Author):

My main comment from before was about the process of selecting the EHR variables. The authors' response was that "a selection is needed in order to eliminate such irrelevant factors and also reduce the number of tests performed". I disagree with this. This study is trying to discover what factors are associated with the metagenome, but they are implying that they already know what is relevant. EHR data can often be associated with outcomes in unexpected ways that do not seem biologically relevant. Even something like "health check for drivers license" tells you that the patient is healthy enough, both physically and mentally, to drive and operate a vehicle--for example, the patient is not in the ICU, is not immobile, and is able to see (the authors include cataract and glaucoma as factors). Reducing the number of tests performed is also not required if the computational resources are available. I would have preferred if this study looked at all EHR variables and showed that the ones that were statistically significant also aligned with the literature. Instead, the authors took diagnoses that they thought were relevant and used this study to confirm which ones really are. They missed the opportunity to discover something unexpected. I still think this is a really interesting paper with important results. I'm just pointing out that there might much more to learn from the amazing database the authors used.

We thank the Reviewer for this comment. We agree with the Reviewer that EHRs data provide an opportunity for more in-depth analysis and discovery of novel links between the microbiome and health. We will kindly take these recommendations into account in our future studies.

The authors also mention in their response that their “aim was to obtain as reliable diagnoses from the EHR as possible”, referencing a 2015 paper by Leitsalu describing the Estonian Biobank. In that paper, Leitsalu defines four levels of “reliability”, with some diagnoses being more reliable if, for example, the diagnosis was both confirmed by the doctor and listed in the database. Does this mean that this new study only used the Class 1 reliable diagnoses from the biobank and ignored the other diagnoses? If so, then that should be stated in the manuscript.

We thank the Reviewer for this important concern. Our health data is indeed most similar to class 1 data, but we do not follow exactly the same classes of reliability as stated by Leitsalu, et al, 2015 as these classes were used during the time when the Estonian Biobank itself did not have access to the Electronic health records. At the beginning of the Estonian Biobank the health data had to be obtained via the family doctors/specialists who were recruiting participants to the biobank and checked their databases for diagnoses. Currently, the Estonian Biobank has access to the national health databases and therefore we do not have to use the doctors to obtain data. In our study, the reliability of the diagnosis data from EHRs was further increased by using at least two data entries from the databases to account for wrong entries or misdiagnoses. Self-reported diagnoses and medications were not included in the analysis. We have now clarified the description of metadata in the Methods section, specifying the selection criteria for the EHRs (pp 19).

Sincerely yours,

Elin Org, PhD

Associate Professor

Estonian Genome Centre, Institute of Genomics

University of Tartu